# Review on the Advancements of Stethoscope Types in Chest Auscultation

**DOI:** 10.3390/diagnostics13091545

**Published:** 2023-04-25

**Authors:** Jun Jie Seah, Jiale Zhao, De Yun Wang, Heow Pueh Lee

**Affiliations:** 1Department of Otolaryngology, Yong Loo Lin School of Medicine, National University of Singapore, Singapore 119228, Singapore; 2Department of Mechanical Engineering, National University of Singapore, Singapore 117575, Singapore; 3Infectious Diseases Translational Research Programme, Yong Loo Lin School of Medicine, National University of Singapore, Singapore 117545, Singapore

**Keywords:** stethoscope, chest auscultation, telemedicine, smart hospital, heart sound, breath sounds

## Abstract

Stethoscopes were originally designed for the auscultation of a patient’s chest for the purpose of listening to lung and heart sounds. These aid medical professionals in their evaluation of the cardiovascular and respiratory systems, as well as in other applications, such as listening to bowel sounds in the gastrointestinal system or assessing for vascular bruits. Listening to internal sounds during chest auscultation aids healthcare professionals in their diagnosis of a patient’s illness. We performed an extensive literature review on the currently available stethoscopes specifically for use in chest auscultation. By understanding the specificities of the different stethoscopes available, healthcare professionals can capitalize on their beneficial features, to serve both clinical and educational purposes. Additionally, the ongoing COVID-19 pandemic has also highlighted the unique application of digital stethoscopes for telemedicine. Thus, the advantages and limitations of digital stethoscopes are reviewed. Lastly, to determine the best available stethoscopes in the healthcare industry, this literature review explored various benchmarking methods that can be used to identify areas of improvement for existing stethoscopes, as well as to serve as a standard for the general comparison of stethoscope quality. The potential use of digital stethoscopes for telemedicine amidst ongoing technological advancements in wearable sensors and modern communication facilities such as 5G are also discussed. Based on the ongoing trend in advancements in wearable technology, telemedicine, and smart hospitals, understanding the benefits and limitations of the digital stethoscope is an essential consideration for potential equipment deployment, especially during the height of the current COVID-19 pandemic and, more importantly, for future healthcare crises when human and resource mobility is restricted.

## 1. Introduction

Stethoscopes were originally designed to listen to a patient’s chest sounds, evaluate the state of the cardiovascular and respiratory system, and determine how well the trachea and bronchial tree are functioning as an airway. It is crucial for medical professionals to listen to the patient’s internal sounds during chest auscultation, especially during the vesicular breath sound periods [1,2,3].

Advancements in healthcare progress dramatically as a result of technological developments, where modern technology enables us to implement multifunctional gadgets with fast processing. Promoting the advancement of technology ensures that devices are becoming more powerful, portable, and convenient and have faster processing speeds than ever before, satisfying the needs of the healthcare industry [4]. For example, using wearable digital stethoscopes for sound recording and visualization offers real-time, wireless, and continuous auscultation via a soft wearable system that can be implemented as a quantitative diagnostic tool for various diseases [5]. The wearable technology can potentially be paired with smartphone applications for continuous auscultation monitoring [6]. These newly designed digital and electronic stethoscopes are technological advancements dedicated to healthcare professionals for use in both clinical and educational settings, thus improving on traditional auscultation techniques [2,7,8].

This review provides a comparison of the different types of stethoscopes to allow for benchmarking, so as to identify potential areas for improvement and to promote effective and efficient use of stethoscopes. It also enables optimum decision making when selecting an appropriate stethoscope for its user, depending on the clinical context.

Figure 1 summarizes the main key points in the advancement of stethoscopes from analogue to electronic and eventually digital stethoscopes. This is further expounded upon in the subsequent sections. Current commercially available stethoscope examples are also shown to illustrate the comparison between the different stethoscope types.

### 1.1. Analogue Stethoscope

Known to be the traditional or conventional stethoscope, the analogue stethoscope consists of a simple mechanical construct for amplifying chest sounds during auscultation [12].

An analogue stethoscope consists of the chest piece and the binaural earpiece, with the bell and diaphragm used for picking up lower- and higher-frequency sounds, respectively [13]. Thereafter, sound waves are propagated via the hollow tubing to the binaural earpiece, providing audio output to both ears [14,15,16]. Takashina et al. also found that adding an ethylene propylene diene monomer as an extensible diaphragm increases the efficiency of sound analysis [17].

Its relative ease of use, cost-effectiveness, and wide availability for purchase by paramedics and nurses makes it a popular choice for healthcare workers [14]. There are, however, limitations with this popular choice. Traditional stethoscopes have varying degrees of sound quality when used on obese patients and patients with thick chest walls, as they tend to have poorer sound quality feedback. Additionally, any circuitry malfunction in the sound pathway can attenuate or eliminate sound transmission, such as an air leak from the hollow tubing or the binaural earpiece [14,15].

In addition, it can only be used by a single operator at any one time, as opposed to its digital and electronic counterparts. It is thus impossible to share, analyze, process, or utilize the sound data in any other manner [18]. This makes it a less viable choice in demonstrating chest auscultation for educational purposes, or for the playback of previous audio readings for cross-referencing when formulating a diagnosis [7]. 

### 1.2. Electronic Stethoscope

The electronic stethoscope has paved the way for a new method of auscultation for research, education, and clinical practice. It allows the amplification of chest sounds through the means of electronic intervention, providing audio feedback, sound level manipulation, audio data recording, and playback [19]. This is made possible via several important processes that occur before sound is fed back at the chest piece and subsequently transmitted to the earpiece [20]. 

By having the chest sounds converted into digital data from analogue sound waves, either via a microphone or a piezoelectric sensor, the transduction of the sound waves into an electrical signal is amplified and processed by passing through bandpass filters that reduce unwanted noise that can corrupt the sound signal. The earpiece would then be able to output real-time amplified chest sound data [14,20]. There is also research on the potential of using higher quality microphones found in smartphones as a substitute for electronic stethoscopes [21]. 

Electronic stethoscopes have surpassed the conventional stethoscope in the sense that the amplification system can be easily used by minimally trained personnel in a primary healthcare setting, and real-time automatic signal amplification allows for immediate patient diagnosis for healthcare workers with hearing impairment [19]. This feature is especially suited for clinicians with hearing loss who may have difficulty hearing softer or subtler sounds. Additionally, electronic stethoscopes make it possible to detect very low-frequency components that conventional stethoscopes cannot detect, making them important for certain diagnoses [22,23]. Certain cardiac murmurs, such as mitral stenosis, especially low-grade ones, are notoriously difficult to pick up on auscultation with a traditional stethoscope alone. Additionally, to remove murmurs at low frequencies, basic filter experiments including the low-pass RC filter, Butterworth filter, Chebyshev filter, and Bessel filter were used to test the signal conditioning quality [24]. Research was also carried out on bedside nursing in a clinical setting to determine if there was a significant difference in the diagnostic utility of the devices, and the electronic stethoscope was found to have superior sound quality in 65% of patients [25].

There are, however, many limitations to this device and its auscultation methods. While independent clinical studies that made use of an electronic stethoscope showed promising results in its applications, they were conducted in an ideal setting, with little to no ambient noise. In contrast, in a realistic clinical setting, background noise and electromagnetic field disruption could easily corrupt the recorded sound data used for electronic amplification, where unwanted signals are amplified. The results of the studies are less reliable as they are not a genuine reflection the real-world scenario, which may limit their utility in a clinical setting [14,26]. On the other hand, Cain et al. tested stethoscopes in simulated helicopter noise at 70 to 100 dB to determine the detection threshold of a normal heart and breath sound [27]. They were unable to detect physiological sounds in the loud noise environment, implying that the ambient noise was amplified after entering the sensing head. The threshold of noise for the detection of heart and breath sounds were 85 dB and 75–80 dB, respectively, indicating the need for at least a 30 dB improvement in the signal-to-noise ratio for it to function properly. To overcome this limitation, there is the possibility of implementing a noise-cancelling feature to counter external background noise so as to aid audibility in noisy conditions such as in disaster zones, chaotic situations, and emergency room scenarios. Tourtier et al. also found that the ability to perform auscultation during air medical transport is compromised by high ambient noise levels [28]. In the context of auscultation during medical transport, the hearing of cardiac sounds was clearer in the electronic amplified stethoscope, but there was no significant difference when the auscultation of breath sounds was considered.

Secondly, the use of electronic stethoscopes requires the approval of the Food and Drug Administration (FDA) and other health professionals’ legalities in other countries for healthcare safety and device maintenance. This hinders the implementation of the device for healthcare workers [14].

Lastly, cost and device availability are major limitations [29]. Developing countries with high patient counts and financial insecurity make it difficult for standard clinics to amass the financial capital to invest in high-quality electronic stethoscopes when there are cheaper and more accurate alternatives currently available [14]. Additionally, the accuracy of the electronic stethoscope may be compromised depending on the manufacturing and mass production processes [30]. 

There can be improvements to the existing electronic stethoscope design to increase the likelihood of implementation. They include designing the device system such that it can be easily used by minimally trained personnel in a primary healthcare setting and provides a simultaneous recording of heart sound data at multiple chest sites with multiple sensors. This may improve diagnostic accuracy since the results from a single heart sound signal can be cross-referenced with those obtained from other sites [19].

Research by Rennoll et al. found that the acoustical characteristics of electronic and acoustic stethoscopes can differ significantly [31]. To test their theory, they introduced a barrier for clinicians to transition to electronic stethoscopes. The study proved that the detection of differences between stethoscopes was significantly less than transitions between the unfiltered electronic and acoustic stethoscope sections, demonstrating the effectiveness of the method used to filter electronic stethoscopes to mimic acoustic stethoscopes, upholding the quality and precision of the medical equipment’s gold standards. 

Overall, the current electronic stethoscope provides a much more advanced and modern solution to overcome the limitations of the analogue stethoscope, as it manages to increase its accuracy and clarity via the amplification of internal body sounds. Devices such as these could significantly assist medical professionals in making a more accurate diagnosis, though the benefits that they offer depend on the process of implementation and design [22].

### 1.3. Digital Stethoscope

Implementing smartphone applications and data processing in the digital stethoscope pushes the boundaries of chest auscultation methodologies. Designed similarly to the electronic stethoscope, it includes a digital filter to remove noise from the electrical signal and extract the signal of interest from the frequency band. Heart and breath sounds obtained in the process are normalized to a particular scale and are segmented into cycles, thus helping in the detection and distinction of the breath and heartbeat components [14,23,26].

Thereafter, the digital data obtained as an electrical signal are further represented in parametric form and are classified as either heart waves or breath sounds. This could assist in making the diagnosis more accurate and improve clinical decision making. The added feature would improve convenience, where audio data obtained from the digital stethoscope are transmitted via Bluetooth to a laptop or a smartphone and analyzed by a pre-programmed software application. Sound waves can be converted into digital data and visually represented on an oscillogram or a spectral graph [14,32]. A smart digital stethoscope system to monitor patients’ heart sounds and diagnose abnormalities in a real-time manner, consisting of two subsystems that communicate wirelessly via Bluetooth low-energy technology with a portable digital stethoscope subsystem and a computer-based decision-making subsystem, was successfully tested for the probability of implementation [33].

Additionally, the use of machine learning or artificial intelligence (AI) incorporated into digital stethoscope applications improves the experience, quality of diagnosis, and care for both patients and healthcare professionals [34]. Alqudah et al. compared different deep learning models for the detection of respiratory pathologies from unprocessed lung auscultation sounds [35]. Their performance suggested that the proposed sets of deep learning methods were successful and achieved high performance in classifying the unprocessed lung sounds. 

DeGroff et al. investigated the neural network-based method for respiratory sound analysis and lung disease detection, so as to automatically deduce the irregularities present in the early stages of a lung pathology [36]. Additionally, by directly gathering the feature vector from breath audio and utilizing supervised machine learning techniques, the algorithm detects unique feature vectors related to a patient affected by lung disease and assists in diagnosing patients based on quantifiable data [37]. Computer-generated machine learning has the potential to be used in analyzing respiratory sounds, similarly to a doctor analyzing breath sounds during auscultation. Another similar example involves research on feature extraction for machine learning-based crackle detection in lung sounds recorded using a stethoscope in a large-scale health survey. The program is trained and used to evaluate audio recordings of crackles classified beforehand by experienced medical professionals. By evaluating several feature extraction methods and classifiers, cross-validation was conducted, where training sets were shuffled between cycles, resulting in precision of 86% and recall of 84% when classifying a crackle in a window [38]. It is potentially well suited for training medical doctors and for the deployment of smart devices, as well as for incorporating machine learning in a clinical setting for aided diagnosis [39].

A huge potential impact in improving the digital stethoscope field includes near-field coherent sensing. The patient’s internal and exterior mechanical movements may be directly modulated onto multiplexed radiofrequency signals combined with a distinct digital identity similar to a digital stethoscope system. The method, which avoids skin-to-skin contact, can monitor numerous people at once and potentially result in the efficient automation of vital signs monitoring in healthcare institutions. This can provide information on heart rate, blood pressure, and respiratory rate without having to make direct contact with the patient’s skin [40].

Digital stethoscopes can process auscultatory heart or lung sound signals, as well as analyze and clarify the resulting sounds, hence assisting in diagnosis based on quantifiable medical assessment and audio data acquisition. Implementing digital stethoscopes in a clinical setting would aid in the automatic interpretation of heart sounds in the realm of cardiovascular diagnostics [19,26,41].

Digital stethoscopes can also be used for the remote monitoring of COVID-19 quarantine patients or treatment of cancer patients who are isolated due to penetrating radiation, such as after the administration of Iodine-131. By giving instructions on the placement of the stethoscope diaphragm, doctors can listen to and analyze the patient’s internal body sounds multiple times with advanced playback and speed-changing options [22,42]. 

Zhang et al. performed a cross-sectional, observational, single-center study on 30 consecutive hospitalized patients with confirmed SARS-CoV-2 pneumonia at Lei Shen Shan Hospital in Wuhan, China [43]. They collected auscultatory characteristics and clinical information on SARS-CoV-2 pneumonia using a wireless stethoscope, which exemplifies how such technology can be practically utilized in a real-world clinical and research setting. Additionally, digital stethoscopes can overcome the pre-existing lack of inter-observer reliability if users are inexperienced or have hearing disabilities. Furthermore, integrating digital stethoscopes with AI could improve the reliability in detecting abnormal breath sounds. Research has shown that developed AI systems can detect crackles and wheezes with a reasonably high degree of accuracy from breath sounds obtained from different existing digital stethoscope devices [30]. Similar research was described by Brunese et al., allowing medical professionals the opportunity to cross-reference patient diagnosis with quantifiable data and audio data analysis [37]. Using an acoustic evaluation methodology based on the Gaussian Mixed Models could assist in the broader analysis, identification, and diagnosis of asthma based on the frequency domain analysis of wheezing and crackles [44]. Similarly, research performed on a microwave stethoscope (MiSt) also demonstrated reliable monitoring of multiple vital signs, such as the respiratory rate, heart rate, and changes in lung water content, through a single microwave measurement [45].

There are also many other benefits to the use of digital stethoscopes in an educational setting. A cardiology patient simulator developed through the application of new digital computer technology is capable of playing back selected physical findings such as cardiac and respiratory sounds that have been pre-recorded from actual patients using digital stethoscopes, and can be configured to a loudspeaker, enabling multiple listeners to listen to the same internal body sounds simultaneously, with on-demand playback [17]. The signal can also be graphically represented on spectrograms, which serve as a visual aid to augment the teaching of auscultation, and also allow analysts to ascribe visual correlations to the exceptionally low-frequency components of cardiac sounds [2,22]. Adventitious lung sounds, such as wheezes and crackles, tend to show recognizable patterns on the spectrogram, which may aid in formulating a clinical impression and in the classification of diseases such as Chronic Obstructive Pulmonary Disease (COPD), as well as enhancing medical students’ learning [46]. Recording of lung sounds may enable the development of audio recording datasets at various vantage points [47]. Certain cardiac diseases have recognizable patterns on an electrocardiogram based on particular electrical conduction abnormalities. By extension, there is potential in exploring abnormal breath sound patterns or waveforms that are pathognomonic of certain diseases, which can be compiled and studied. Rice studied the methodologies of imparting auscultation skills to students based on verbal articulation and sensory knowledge [48], while Ferns & West developed a practical guide to auscultation techniques for the respiratory system for nurses developing advanced practice skills [49].

Figure 2 summarizes the main characteristics and features of digital stethoscopes that make them potentially applicable in a wide variety of clinical settings amidst the rapid pace of technological advancements. 

Accuracy and Precision: Given that sound waves are generated and can be recorded, it would remove the element of human error when assessing for heart or lung sounds, and the subsequent interpretation of such sounds. This would be even more beneficial if there were a database of recordings that has been pre-interpreted and correlated to a disease pathology, and if AI or machine learning were able to match these sounds to the recordings in the database. 

Reproducibility: The audio sensor would technically be able to consistently give accurate readings regardless of the situation or context in which the readings are being recorded. 

Ambulatory Assessment: Heart sounds can be measured beat-to-beat for a designated period of time, without the patient having to remain in the healthcare facility. Moreover, it is also a more physiological representation of how a patient goes about his or her daily life, such as during periods of high exertion when exercising, as opposed to periods of rest. If a certain wireless sound sensor can be feasibly attached onto the patient’s chest wall, recordings can be remotely saved into a database, which would allow for subsequent analysis by an expert such as a cardiologist. 

Simultaneous Assessment: Audio sensors can be placed simultaneously in several locations in the body. For example, sensors can be placed at different locations of the heart, as well as placed on different locations of the lung fields, allowing both breath and heart sounds to be assessed at the same time and correlated to each other. Mechanical or thermal sensors can also be implemented to correlate findings, such as when certain heart sounds are magnified during specific parts of the breath cycle. Other potential applications would be to simultaneously monitor and record vital signs and correlate them to the findings of the digital stethoscope. 

Remote Accessibility: This would allow a clinician who is not physically present to listen to the patient’s heart or breath sounds. This could be useful in a certain number of scenarios, such as poor access to healthcare due to geographical distance, requirements for an urgent specialist opinion, convenience for both patient and healthcare professional, as well as for educational purposes, where many students can listen to rare heart or lung sounds that they would otherwise not have been able to without seeing the patient. 

Sound analysis and Modification: Digital stethoscopes can assist clinicians with hearing loss by amplifying the volume of certain frequencies that they would not have been able to hear otherwise, which may reduce missed diagnoses. A noise-cancelling feature can be added to avoid inaccuracies in interpreting sounds due to background noise, when out in the field, in disaster areas, or in a crowded clinic or emergency department. 

Recordability: Heart or breath sounds can be recorded and saved for future educational purposes. It can also be applied in medicolegal cases, such as if there is a dispute in the assessment of a patient at a particular point of time, where the recording of sounds will serve as permanent and objective documentation of the patient’s disease state. 

Longitudinal Assessment: The patient’s heart or breath sounds can be charted over a period of time, to assess whether his or her disease course is improving, worsening, or remains stable. For example, patients with interstitial lung disease frequently have bilateral fine crepitations on auscultation. However, over time, the severity, volume, and character of these crackles may change and evolve, giving a clue to the disease course, suggesting the possible onset of a new disease, or prompting revision of the diagnosis.

While some of the limitations of the electronic stethoscope have been overcome through the means of developing new features and attributes in its digital counterpart, some obstacles and other limitations have also been identified.

Firstly, heart sounds and murmurs have complex characteristics that are dynamic in nature. The same disease can have different types of murmurs and different diseases can have similar murmurs. The murmur can even vary within the same patient, with the onset of arrhythmia or cardiac dysfunction affecting auscultatory findings. With this in mind, AI may not be able to classify them as accurately as expected. This would then require the development of more complex algorithms and the use of deeper machine learning to overcome the issue [14]. 

The solution to an accurate diagnosis may also be improved with inputs from other sensors. Park et al. recently proposed a multimodal technology-based smart stethoscope prototype for personal cardiovascular health monitoring [50]. Installed with a digital microphone and photoplethysmography sensor, information regarding heart sounds and pulse rate can be simultaneously obtained and wirelessly transmitted via a smartphone to diagnose cardiovascular disease. Diagnostic accuracy is increased due to simultaneous multi-sensor data comparison capabilities and cardiovascular disease self-diagnosis algorithms.

Similarly, there is research on cutting-edge medical equipment with additional capabilities on top of existing designs of the digital stethoscope capsule. Medical equipment exchanges data and information across great distances due to wired and wireless connectivity. The encapsulation features will allow for the capture and transmission of electrocardiographic activity through Bluetooth, determined at the same cardiac focus and synchronized with the phono-cardio graphic sound recordings, using low-cost hardware technologies. Utilizing 3D printing, many encapsulation solutions were prototyped for attachment to the digital stethoscope, and subsequently tested for comfort, the collection of acoustic and electric signals, and ergonomics in a hospital setting. The final prototype’s performance was up to clinical standards [51].

Secondly, variations in the datasets used in studies may be due to differences in population characteristics and ethnicity, hence posing a potential limitation. Thus, algorithm validation would need to be performed with larger datasets, with appropriately blinded trials to reduce bias [14]. Large-scale crowdsourced datasets of respiratory sounds have been analyzed to aid the diagnosis of COVID-19. A study by Brown et al. described a simple binary machine learning classifier that can utilize cough and breathing sounds to understand and discern those from COVID-19 against sounds attributable to asthma or healthy controls [52].

Lastly, there is difficulty for older and more experienced healthcare professionals to integrate and transition to the newer and more complex digital stethoscope systems, as they are accustomed to the traditional stethoscopes, and they may require multiple medical device training sessions to remain updated [46,53].

### 1.4. Potential Barriers to Implementation

While there may be numerous benefits conferred by technological advancements in stethoscope quality, clinicians may still opt for the use of the traditional stethoscope for a myriad of reasons. These barriers to the implementation of the aforementioned advanced stethoscopes present an obstacle in how clinicians transition to and accept the newer technology, and must be addressed to reap the potential benefits. 

The main challenges that arise from the use of such advanced stethoscopes include higher costs, not only during the initial purchase but also for the repair or replacement of damaged or misplaced devices. Costs increase in parallel with the increasing sophistication of the stethoscope. Moreover, as more features and parts are incorporated to improve the stethoscope, reduced ergonomics may be a potential drawback. Advanced stethoscopes may be heavier and more bulky, which could prompt clinicians to favor the comparatively lightweight and portable traditional stethoscope. The learning curve faced when transitioning to an advanced stethoscope may also discourage clinicians from abandoning the traditional stethoscope that they have been accustomed to for many years, especially among those who are more senior or experienced. More importantly, potential suboptimal usage of new technology during the initial transition period may lead to misdiagnoses, which can be detrimental to patient care. Last but not least, with more sensitive and sophisticated devices, certain bodily sounds may be picked up by such advanced stethoscopes, but with unknown clinical significance; there is potential for overinvestigation or undue patient anxiety which causes more harm than benefit. With the documentation of heart or lung sounds that comes with the ability to record sounds, the medicolegal standard may be raised, which can potentially create unnecessary defensiveness in clinical practice, overall diminishing patient utility. 

While these issues may inevitably exist, targeting the clinician’s mindset and the provision of accessible resources may mitigate the barriers to implementation. With increased research and evidence on the benefits of advanced stethoscopes being made available, it may help to encourage the transition towards a modern tool that can potentially improve patient outcomes.

## 2. Benchmarking Methods

To better understand the quality and features of the currently available stethoscopes in the healthcare industry, providing a comparison of the different stethoscopes would allow for benchmarking. Areas of improvement in the stethoscopes can be identified, enabling optimum decision making when selecting an appropriate stethoscope for its user, as well as for continued innovation and technological advancement. Figure 3 conceptualizes the main methods in which the benchmarking of stethoscopes is performed, and this is further discussed in the subsequent sections. 

### 2.1. Human-Based Comparison

Benchmarking of stethoscopes usually involves reliance on experienced healthcare professionals as a quality assessment indicator. A common method would be to compare the assessment feedback from stethoscope users at different levels of clinical experience in auscultation [15,54]. 

A clinician can use two different types of stethoscopes to auscultate the same patient twice and determine whether abnormal heart and lung sounds can be correctly identified. These findings could aid in determining clinical feasibility and equipment inferiority [54]. Similarly, clinical personnel with different levels of experience can be engaged to cross-reference data accurately [7]. The investigators could also measure inter- and intra-observer agreement with the Kappa index via a questionnaire. By doing so, the performance between subgroups of doctors based on clinical experience would provide the estimated proportion difference of correctly identified heart and lung sounds for each subgroup [54]. Another unique example of medical professionals’ benchmark data quality would be the detection of murmurs and gallops identified in cats with heart disease. However, auscultatory findings may be subject to clinically relevant observer variation when comparing electronic to conventional stethoscopes in the detection of abnormal heart sounds [55].

### 2.2. Audio Recording Data Comparison

Audio data from chest auscultation could be collected using different stethoscopes, analyzed, and compared against each other to benchmark the sensitivity and precision of the reading [42]. A similar study by Rappaport & Sprague was performed, but also tested for the improper fitting of stethoscopes in the ear [56].

Auscultatory sounds using an electric microphone placed within one earpiece could be recorded, with the other earpiece sealed. Each type of stethoscope is equipped with a different type of electroacoustic transducer for the conversion of vibrations of the underlying skin into an electric signal. The stethoscopes also have a set of selectable digital filters and different frequency range modules for reading certain audio waves [7,42]. Liu et al. found that with regard to physiological mechano-acoustic signals, often the frequencies and intensities beyond the audible range provide information of great clinical utility [57]. Thereafter, auscultatory data points were collected and verified with an echocardiogram. The analysis showed that the sensitivity in detecting valvular regurgitation was higher when using the electronic stethoscope as compared to the traditional stethoscope, indicating a significant increase in sensitivity conferred by the electronic stethoscope. However, the specificity of audio data was equally high for the electronic and acoustic stethoscopes [7]. The creation of a computer-based program that analyzes respiratory sounds automatically may potentially be used for telemedicine and self-screening [58,59].

### 2.3. Feature-Based Benchmarking

Another method of comparison involves differentiating the stethoscopes based on their unique specifications and features. The physical and pre-programmed user-designed traits aid in benchmarking the limits of certain stethoscope design capabilities, and the resulting comparison of the advantages and disadvantages allow for more informed decision making when selecting an appropriate stethoscope [14]. Similarly, there is research that tested six different analogue stethoscopes for their quality and accuracy [60].

Another example would be comparing the specification feature limits of different electronic stethoscopes. By understanding the frequency bands of certain stethoscopes, we can tailor the feature selections to specific patient requirements. As infants have lung sounds containing higher-frequency components as compared to adults, auscultation of the infant’s chest would be different from that of adults [61,62,63]. Research by Ramanathan et al. has also shown the effectiveness of the technical properties and how the limitations of stethoscope technology could affect potential uses in the fields of pediatrics and neonatology, from telemedicine to computer-aided diagnostics [64]. Studies were also performed to provide non-invasive methods to monitor the heart rate and respiratory rate in neonates [65] and to assist in bedside monitoring and the diagnosis of pediatric and cardiology patients [66].

### 2.4. AI and Audio Data Comparison Analysis

Another means of comparison would be through AI software that analyzes the collected audio data from digital stethoscopes and makes diagnoses based on presented measurable symptoms, and subsequently compares the results to those of an experienced healthcare worker using a traditional stethoscope. By doing so, we can compare and justify the diagnosis of patients based on representable audio data sound quality and confirmed data acquisition. For example, normal or abnormal heart and breath sounds can be recorded and serve as a formal documentation of the patient’s disease state at a particular point of time, and, in the event of medicolegal disputes, this can present valuable information about the patient’s disease state retrospectively.

An example would be collecting multiple auscultatory recordings from patients using two different digital stethoscopes, where each patient is classified and labelled by an experienced pediatric respiratory physician as containing wheezes, crackles, or neither. Based on audio playback and careful spectrogram and waveform analysis, with subset validation by a blinded second clinician, such recordings were submitted for analysis by a blinded AI algorithm specifically trained to detect pathologic pediatric breath sounds [30]. The crackle detection positive percentage agreement showed promising results with the optimized AI detection thresholds, allowing the AI to detect crackles and wheezes with a reasonably high degree of accuracy based on breath sounds obtained from different digital stethoscope devices. The analysis of different deep learning models suggested that all the proposed deep learning methods were successful and achieved high performance in classifying the unprocessed lung sounds [35,38]. Similarly, there is research on the use of embedded stethoscopes designed to serve as a platform for the computer-aided diagnosis of cardiac sounds for the detection of cardiac murmurs [67], with other research advancing to a portable device with the capability to diagnose cardiac pathology in real time, employing the signal conversion of analogue acoustic signals into a digital signal that can simultaneously be displayed on a computer using a MATLAB graphic user interface for visual representation, thereby enabling a critical analysis of the interpreted data [68]. This can be used as a clinical tool for the diagnosis of valvular and other structural heart diseases in educational settings [69]. Another example would be the use of a conventional stethoscope with a condenser microphone built into the head to record cardiopulmonary sounds and an AI-based classifier for these sounds. Fast Fourier transform (FFT) analysis was used to examine and assess various stethoscope head microphone deployments with amplification and filter circuits to assess the impact of noise reduction quality [70]. Zhang et al. studied breath sounds collected by two experienced pediatric pulmonologists and six general pediatricians from the respiratory department of Shanghai Children’s Medical Center by using an electronic stethoscope, and the accuracy, sensitivity, specificity, precision, and F1-score of the AI algorithm were determined [71]. It was found that the ability of the AI algorithm to analyze adventitious breath sounds was better than that of the general pediatricians.

These serves to show that the ability of digital stethoscopes paired with AI programming systems could uphold quality, in terms of accuracy and precision in diagnosis based on quantifiable symptoms, improving and aiding the overall diagnostic ability of experienced medical personnel with a digital stethoscope.

## 3. Potential Applications and Implementation of Digital Stethoscopes

### 3.1. Notable Advancements in Telemedicine

Even before the COVID-19 pandemic, telemedicine catered to numerous medical specialties to make medical care more accessible, improve healthcare infrastructure to encourage healthcare communication, alleviate manpower shortages, and ease patient monitoring. Bridging access to pulmonology specialist care, rehabilitation, symptom monitoring, and early identification of clinical exacerbations, the use of telemedicine has aided intensive care settings, and has improved patient outcomes and addressed manpower issues due to the increase in the monitoring frequency required for critically ill patients amidst the COVID-19 pandemic [72]. The digital stethoscope can be implemented to overcome the limitations of a conventional or electronic stethoscope as the sound data are transformed into electrical signals, which can be amplified, stored, played back, and transmitted across long distances to a medical expert, making it very useful in telemedicine [73]. Nurses and doctors may use the medical information obtained by electronic stethoscopes in both telemedicine and face-to-face clinical settings for teaching [74]. Advanced research studies have been performed, which monitor the degree of success in certain countries such as Canada’s telehealth programs [75], and Singapore’s postnatal care with telehealth [76]. Another example of a telehealth study in Singapore explores the potential use of telehealth in an orthopedic clinical setting including HIPAA-compliant peer-to-peer communication, with clinical outreach in the setting of trauma [77]. It provides both patient and parent reassurance in pediatric orthopedic patients, with excellent results in the integration of a secure mobile telehealth application and messaging platform. Similarly, there is a research on the recent conversion to telehealth options among Singapore’s dialysis centers due to the rise in COVID-19 cases, with integrated computerized physicians and online telephone discussions with dialysis nurses, which was noted to enable good public engagement and yield high patient satisfaction scores [78].

#### 3.1.1. Benefits

One benefit of telemedicine is the tele-management of chronic respiratory disorders—in particular, conducting online consultations and monitoring of pediatric asthma patients using digital stethoscopes and otoscopes. It was proven that it provided equivalent service and care to face-to-face evaluations [72]. Similarly, according to new research on asthma by Portnoy et al., telemedicine has been found to work well [79].

Another example would be the integration of telemedicine into the general public’s healthcare to increase patient access to specialty care and symptom monitoring in dense pediatric populations such as schools. Current research assessing the impact of telemedicine on absence from childcare due to illness holds substantial potential to reduce the impact of illness on the health and educational development of children [80]. Additional studies analyzed the quality and cost-effectiveness of healthcare provided in urban and rural elementary school-based telehealth centers [81]. The telemedicine platform allowed for remote asthma management comprising smartphone-based teleconsultation with nurses and wireless symptom tracking with a digital stethoscope. The main benefit was noted to be the management of asthmatic conditions, providing an irreplaceable medium to monitor patients offsite by instructing on inhaler techniques, assessing airway compliance, and providing personalized asthma care consultations for immunocompromised children [79]. Simeone et al. found that treatment plans in 22.4% of asthma patients who had unreliable in-person consultation follow-up appointments due to COVID-19 were beneficially altered [72]. This is also further elaborated in a study by Jain et al., on the reliability of follow-up care with telemedicine aid and treatment plans [82].

Telemedicine can also prove beneficial during disaster responses. Disasters have a wide range of negative repercussions, resulting in times of stress for healthcare systems from a medical standpoint. Telemedicine can improve access to medical care during emergencies and reduces stress on medical facilities [83,84]. The practice of telemedicine has also significantly increased as a result of developments in communications technology and market variables used in disaster relief as a resource for first responders, as well as being a direct line for patients [85]. Pasipanodya & Shem studied patients who suffered from spinal cord damage and received outpatient telemedicine before departure from acute inpatient rehabilitation to address urgent and ongoing medical issues, as well as to improve the replenishment of medical supplies during times of crisis [86]. Emergency planning that takes disability needs into account is essential to satisfy specific medical requirements and reduce the impact of disasters on medical professionals, making telemedicine a useful tool. However, telehealth relies on network connectivity to operate, which may be unavailable during times of disaster or network disturbances. A review by Almathami et al. noted that the internet speed had an impact on how well home online health consultations worked and, in 75% of cases, stated that participants were dissatisfied with the poor video and audio quality, resulting in negative feedback on the medical consultation [87].

#### 3.1.2. Limitations

Although telemedicine has demonstrated benefits in providing remote clinical management and the consistent monitoring of patients with chronic respiratory diseases, some studies have found that these benefits may not extend to all patient demographics. 

In the treatment of COPD, several quality-of-life indices (Clinical COPD Questionnaire; CCQ) were compared between patients monitored via telemedicine and conventional office-based encounters. Telemedicine was associated with significantly lower CCQ scores and significantly higher pulmonology follow-up appointments [72], and this was also proven by research on COPD medical care quality [88]. Electronic stethoscope integration demonstrated optimal agreement between face-to-face auscultations and virtual examinations, enabling providers to perform real-time virtual pulmonary examination and assessments. The wide-scale application of this technology is currently limited by the cost and the logistics of deploying electronic stethoscopes into communities with fewer resources, such as those affected by poverty [72,89]. The 2015 Chennai floods in India illustrates a real-life example of the limitations in telemedicine support during times of disaster, which noted the reliance on electricity and network coverage. The more connected the area was to electricity, the better the access to aid. However, the communities that required the most assistance had little to no connection to electricity and were neglected for a longer period [90].

### 3.2. Potential Advancement in Wearable Devices Paired with Digital Stethoscopes

Currently, digital stethoscopes and advancements in wearable sensor devices are implemented in non-invasive sound acquisition hardware devices and include techniques for remote respiratory and sound acquisition. These sounds are then processed and analyzed for clinical, scientific, or educational purposes [91].

Using wearable digital stethoscopes for sound recording and visualization offers real-time, wireless, continuous auscultation via a soft wearable system as a quantitative diagnostic tool [5]. The wearable technology could potentially be paired with smartphone applications for continuous auscultation monitoring [6]. 

Chen et al. studied a small, potentially low-cost, and wearable piezoelectric heart sound sensor, which is suitable for long-term dynamic monitoring and provides technical support for the preliminary diagnosis of heart diseases [92]. Gupta et al. explored the construction of a micro-sensor to capture the body’s mechano-acoustic signals in a wide frequency range to enable the longitudinal study and monitoring of the cardiopulmonary system, detecting a wide range of vibrations on human skin [93]. This ranges from very low-frequency movements associated with the chest wall and body position, to high-frequency acoustic signals produced by the heart and lungs. It could also be implemented in a sleep study to detect sleep-disordered breathing and COPD [5]. Yilmaz et al. mentioned the potential of a sound acquisition module integrated into a dedicated garment worn by the patient, thereby minimizing the role of the patient in positioning the stethoscope and applying the appropriate pressure [94]. The study benchmarked the device against an electronic stethoscope widely used in clinical practice to quantify the assessment.

Klum et al. mentions a possible combination of sensors and prototypes that are non-invasive, wearable, and Bluetooth 5.0 LE enabled [95]. This multimodal sensor patch combines micro-electromechanical systems (MEMS) stethoscope function, ambient noise sensing, electrocardiography, impedance pneumography, and 9-axial actigraphy, reducing the need for sensors at different body positions, allowing for cross-data comparison and enables long-term auscultation. Lee et al. described a biometric system based on cardiac sound authentication using a soft, wearable digital stethoscope offering a continuous security system for the users [5]. The individual heart sounds were collected using the device, and after undergoing two-stage filtering, unique waveforms were achieved for individuals for the Convolutional Neural Network (CNN)-based machine learning model. It was observed that the accuracy of the CNN classifier was 98.3%, making it a very reliable security measure for biometric security studies.

### 3.3. Contributions to Innovation of Smart Hospitals

Smart hospitals can increase efficiency, enhance treatment quality, and open up access to more people with rising healthcare demands and costs. Therefore, hospitals aim to use technology and data to streamline procedures when managing physical assets, exchanging information, and assisting staff in providing optimal care to patients [96]. Smart hospitals can also have an impact on health and medical policy, as well as provide new medical value [96]. Examples include using short-range communication technology or high-speed communication network-based services based on new wireless communication technology, location recognition, and tracking technology, which are used to monitor information regarding the position of an object [97]. 

Digital stethoscopes can contribute to smart hospitals in terms of checking the heart rate, aid regular check-ins with numerous doctors, and assist in the precise planning required to manage rare conditions [98]. Remote patient monitoring with digital stethoscopes can also help them stay ahead of treatment through telehealth, such as in surveillance, contact tracing, diagnosis, management, and prevention amidst the COVID-19 pandemic, ensuring that healthcare is delivered effectively. These highlight the common uses of AI, telemedicine, and digital platforms in smart hospital systems [99].

### 3.4. Preventive Diagnosis and Monitoring

Stethoscopes can be used as non-invasive options for diagnosis and monitoring. Katarzyna showed that it can be used as a cardiac screening tool for preoperative evaluation of patients prepared for hip replacement surgery [100]. Similarly, in areas with limited resources, an electronic stethoscope with an integrated phonocardiogram might be a valuable tool for pediatric heart examination [101]. 

Additionally, potential diagnoses of chronic illnesses found in the joints of elderly patients can be examined using acoustic wave technology directed at the knees. The essential signals are gathered using an electronic stethoscope or high-resolution recording equipment so that the wide-frequency audio signals of the knee joint may be measured. The stethoscopes aid the process of distinguishing between those who have healthy as opposed to degenerative knee joints, which would allow doctors to choose the best rehabilitation techniques [102]. King et al. highlighted its monitoring function, such as using the digital stethoscope to assess lung aeration in neonatal respiratory distress syndrome [103]. The electronic stethoscope could also be used to detect coronary artery disease that produces weak murmurs, which might be detected via analysis with an applied method for diagnosis [41]. AI-based algorithms can also be used to identify aortic stenosis with the help of digital stethoscopes [104]. If low-grade murmurs can be picked up early, it could lead to the earlier detection of a disease that would have been otherwise clinically asymptomatic. The necessary workup can subsequently be performed, leading to the earlier detection of the disease in its pre-clinical stage and also a potentially longer treatment window, hopefully leading to improved clinical outcomes. 

Research pertaining to stethoscopes for cervical auscultation, typically performed with an analogue stethoscope, have also advanced due to the implementation of digital stethoscopes, allowing for more accurate and improved signal data [105]. Ma et al. developed an improved bi-ResNet deep learning architecture, called LungBRN, which uses Short-Time Fourier Transform (STFT) and wavelet feature extraction techniques to improve the diagnostic accuracy of auscultation for medically under-served populations with critical illnesses [106]. They were able to apply sound classification through a digital stethoscope to provide an immediate diagnosis for COPD. 

## 4. Summary and Outlook

Based on the ongoing trends in the advancement of wearable technology, telemedicine, and smart hospitals, understanding the benefits and limitations of the digital stethoscope is an essential consideration for potential equipment deployment and technological implementation. This is especially so amidst the current COVID-19 pandemic, and more importantly for future global health crises, where human and resource mobility could be limited. Essential medical consultations and real-time monitoring of patients can be performed without being limited by geographical boundaries, making available a potentially beneficial lifeline during times of disaster. Telemedicine with the use of digital stethoscopes will ultimately still require a reliable supply of electricity and communication network facilities. These requirements can, however, be unfulfilled in situations such as severe weather conditions, natural disasters, or war. 

There are many methods available to perform a comparison of the different types of stethoscopes for the purpose of benchmarking, allowing for improved credibility and result validation, potentially creating further research opportunities for the healthcare industry.

Healthcare technology improves continuously over time, overcoming the limitations of the previous stethoscope options by integrating new medical engineering solutions into fast-paced clinical and educational settings. Hopefully, the advancement of modern technology will create more accurate and effective medical devices that can be used to perform auscultation to obtain a medical diagnosis, benefiting both patients and healthcare professionals.

## Figures and Tables

**Figure 1 diagnostics-13-01545-f001:**
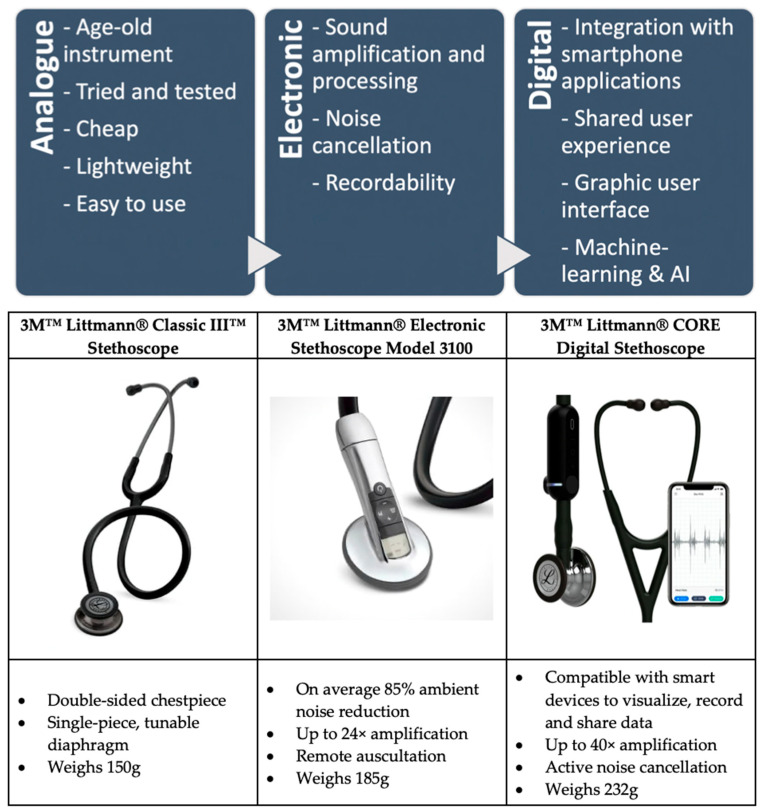
Technological evolution and advancements of stethoscopes over the years, with commercially available examples for cross-comparison [9,10,11].

**Figure 2 diagnostics-13-01545-f002:**
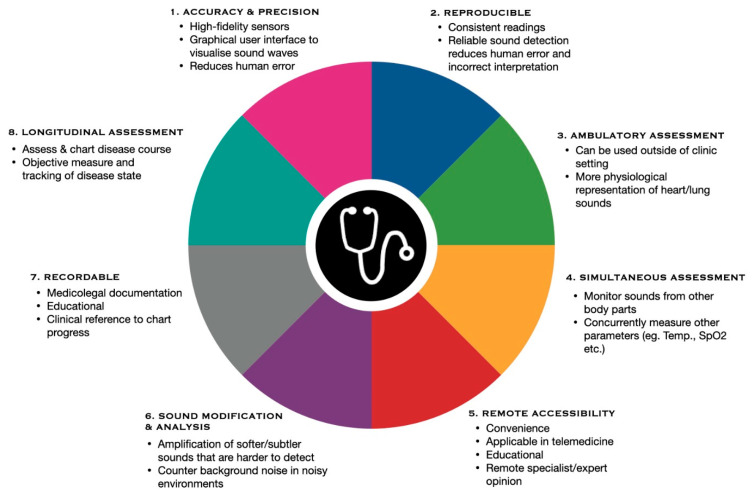
Main characteristics and features of digital stethoscopes that make them potentially attractive in comparison to the traditional stethoscope.

**Figure 3 diagnostics-13-01545-f003:**
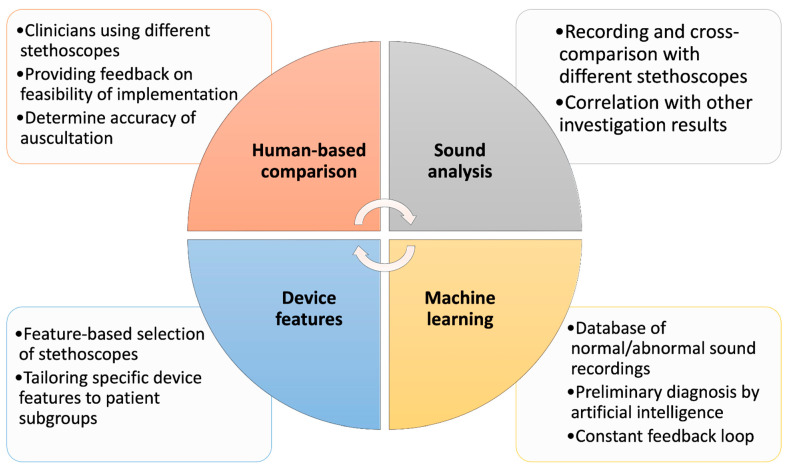
Main domains of benchmarking stethoscope utility in the context of progressive technological innovation.

## Data Availability

Not applicable.

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
