# Peer review of "Review on the Advancements of Stethoscope Types in Chest Auscultation"

_diagnostics, 2023, doi:10.3390/diagnostics13091545_

Round 1

Reviewer 1 Report

Stethoscopes were originally designed for the auscultation of a patient’s chest with the purpose of listening for lung and heart sounds. These aid medical professionals in their evaluation of the cardiovascular and respiratory systems, as well as in other applications such as listening for bowel sounds in the gastrointestinal system, or assessing for vascular bruits, to name a few. Listening to internal sounds during chest auscultation aids healthcare professionals in their diagnosis of a patient's illness.

The authors proposed a  review covering  the current stethoscopes available to healthcare professionals specifically for its use in chest auscultation. With medical devices becoming more efficient than before, diagnostic accuracy is increased and treating patients has become more convenient. Each type of stethoscope comes with a set of unique features and attributes that aid in chest auscultation. By understanding the specificities of different stethoscopes available on the market, healthcare professionals can capitalize on their beneficial features, to serve both clinical and educational purposes. Additionally, the ongoing COVID-19 pandemic has also highlighted the unique application of digital stethoscopes for telemedicine. Thus, the advantages and potential limitations of digital stethoscopes will be reviewed. Lastly, to determine the best available stethoscopes in the healthcare industry, this literature review explores various benchmarking methods that can be used to identify areas of improvement for existing stethoscopes, as well as to serve as a yardstick for the general comparison of stethoscope quality.

The authors concluded that: (a) The potential use of digital stethoscopes for telemedicine amidst ongoing technological advancements in wearable sensors and modern communication facilities such as 5G are also discussed. (b) Based on the ongoing trend in advancements in wearable technology, telemedicine, and smart hospitals, understanding the benefits and limitations of the digital stethoscope is an essential consideration for potential equipment deployment, especially during the height of the current COVID-19 pandemic and more importantly for future healthcare crises when human and resource mobility is restricted.

The study is both stimulating and innovative.

It needs some improvements.

Strengths

Study conducted with enthusiasm

Interesting and well covered topic

Points of weakness

The presentation does not follow a standard structure

More attention to detail is needed (for example the figures)Further comments

1.      The abstract mus better summarize the sections (for example the methods of review)

2.      Figure must must be described in details.

3.      References must be cited with [].

4.      The content is fine. I suggest to rearrange the study inserting the methods/results/discussion/conclusions to improve the message.

Author Response

Dear Reviewer, 

Thank you for your valuable comments. 

We have re-organized our abstract to better signpost the different sections and included the methods of review. In addition, we have also described our figures in greater detail, and also improved on our figures to provide more information. Where applicable, we have also provided more elaboration and clarity in the main text. 

We are grateful for your valuable comments which have helped improve on the quality of our review. 

Reviewer 2 Report

Congratulations on a comprehensive review piece. It reads well. However, some of the concepts are repeated and a trimmed-down version may be worth pursuing if the editors suggest, as there is potential for increased succinctness.

One additional concept that may be worth mentioning is 'barriers to implementation' with clinician reservations / a resistance-to-change mindset being a problem when it comes to altering the way medicine is practiced, even when novel technological tools with evidence are there.

Author Response

Dear Reviewer,

Thank you for your valuable comments.

We have improved on the succinctness of our review and removed some of the repeated concepts as per your suggestion.

We have also included a new section on “barriers to implementation” as suggested.

We appreciate your kind feedback which has helped to improve the quality of our paper.

Reviewer 3 Report

Dear Authors,

Overall your manuscript on the survey of stethoscopes for medical applications reads well. There are some minor points to help improve the readership:

1) Fig 2 and Fig 3 could be moved before in the manuscript. Fig 3 is towards the end and readers may not see this figure at this place. Also, Fig 2 could be re-drawn to make it look more appealing. In fact Fig 3 is more appealing and could be brought before Fig 2. Adding more figures or tables could enhance the appeal as well.

2) The section on wearable devices integrated with stethoscope signals is interest and could be elaborated. There are a number of commercial wearable devices that could be used with stethoscopes. A recent review covers several examples for your guidance: https://onlinelibrary.wiley.com/doi/10.1002/aisy.202100099

3) A table could be provided of stethoscope manufacturers and their equipment with their state-of-art parameters for comparison. Adding some figures compiled from published papers about recent trends in the different sections could be beneficial.  

Author Response

Dear Reviewer,

Thank you for your valuable comments.

We have reordered the positions of Fig 2 and Fig 3 to introduce them earlier in the manuscript as per your suggestion. In addition, we have also re-drew Fig 2 (old, now currently Fig 3 after revision) to make it more appealing. We have added a table to introduce the currently available stethoscope as examples to illustrate the various development of stethoscope features. Where appropriate, we have also elaborated on the various topics as directed.

We are grateful for your suggestions which have improved the quality of our review.

Reviewer 4 Report

I understand the article like a well processed introduction to this topic. To my opinion, a brief comparison of on the market available stethoscopes will improve the value of your article, because the collection of the data will be much more completed.

Author Response

Dear Reviewer,

Thank you for your valuable comments.

We have added a table and integrated it with Fig 1 to aid with comparison of the current market available stethoscopes as examples to illustrate the various development of stethoscope features.

We appreciate your kind feedback which has helped to improve the quality of our paper.

Round 2

Reviewer 1 Report

N/A